# Impact of Iodine Electrodeposition on Nanoporous Carbon Electrode Determined by EQCM, XPS and In Situ Raman Spectroscopy

**DOI:** 10.3390/nano13091545

**Published:** 2023-05-04

**Authors:** Harald Fitzek, Martin Sterrer, Daniel Knez, Horst Schranger, Angelina Sarapulova, Sonia Dsoke, Hartmuth Schroettner, Gerald Kothleitner, Bernhard Gollas, Qamar Abbas

**Affiliations:** 1Graz Centre for Electron Microscopy (ZFE), Steyrergasse 17, 8010 Graz, Austria; 2Institute of Physics, University of Graz, Universitätsplatz 5, 8010 Graz, Austria; 3Institute of Electron Microscopy and Nanoanalysis (FELMI), Graz University of Technology (TU Graz), NAWI Graz, Steyrergasse 17, 8010 Graz, Austria; 4Institute for Chemistry and Technology of Materials, Graz University of Technology, Stremayrgasse 9, 8010 Graz, Austria; 5Institute for Applied Materials (IAM), Karlsruhe Institute of Technology (KIT), Hermann-von-Helmholtz-Platz 1, D-76344 Eggenstein-Leopoldshafen, Germany; 6Helmholtz Institute Ulm for Electrochemical Energy Storage (HIU), Helmholtzstrasse 11, 89081 Ulm, Germany; 7Faculty of Chemical Technology, Institute of Chemistry and Technical Electrochemistry, Poznan University of Technology (PUT), 60965 Poznan, Poland

**Keywords:** batteries, supercapacitors, carbon-electrodes, iodide-electrolytes, conversion-type cathode, hybrid supercapacitors

## Abstract

The charging of nanoporous carbon via electrodeposition of solid iodine from iodide-based electrolyte is an efficient and ecofriendly method to produce battery cathodes. Here, the interactions at the carbon/iodine interface from first contact with the aqueous electrolyte to the electrochemical polarization conditions in a hybrid cell are investigated by a combination of in situ and ex situ methods. EQCM investigations confirm the flushing out of water from the pores during iodine formation at the positive electrode. XPS of the carbon surface shows irreversible oxidation at the initial electrolyte immersion and to a larger extent during the first few charge/discharge cycles. This leads to the creation of functional groups at the surface while further reactive sites are consumed by iodine, causing a kind of passivation during a stable cycling regime. Two sources of carbon electrode structural modifications during iodine formation in the nanopores have been revealed by in situ Raman spectroscopy, (i) charge transfer and (ii) mechanical strain, both causing reversible changes and thus preventing performance deterioration during the long-term cycling of energy storage devices that use iodine-charged carbon electrodes.

## 1. Introduction

To achieve the goal of environmentally friendly and cost-effective electrochemical energy storage, conversion-type electrodes with highly reversible reactions are more suitable than intercalation-type electrodes. Iodine possesses a high capacity of 211 mAh g^−1^ (higher than Li-ion cathode materials, such as LiFePO_4_ = 170 mAh g^−1^), it undergoes a fast iodine/iodide redox reaction, and can be reversibly stored inside the porous network of high surface area carbon, making it an ideal conversion-type cathode [1]. The cyclability of devices assembled with an iodine-carbon cathode is very high at elevated C-rates, up to 6000 cycles for Zinc-iodine batteries and over 100,000 cycles for supercapacitors [2,3,4,5]. Therefore, the impact of iodine storage on the physical and chemical characteristics of the carbon host needs to be investigated. Previous studies on the carbon-iodine interface have been reported from a theoretical point of view [6,7,8] on model systems such as graphene [9,10,11,12,13,14,15], carbon-nanotubes [16,17,18,19], and amorphous carbon [20,21,22]. In addition, ex situ Raman spectroscopy and thermogravimetry have been done on the carbon electrodes after electrochemical treatment in aqueous iodide electrolytes [23,24,25], providing some insight into the ageing of the carbon. However, a detailed assessment of the carbon-iodine interactions at every step from the pristine carbon to the electrified conditions is lacking. The interpretation of the Raman spectra of carbon from a structural [26,27,28,29,30,31,32,33,34,35,36,37,38], and electrochemical [39,40,41,42,43,44,45,46,47,48,49,50,51] point of view, has shown only subtle changes in the D- and G-bands.

This work provides insight into the carbon structure and surface chemistry with a special focus on the reversible and irreversible processes throughout the charge/discharge of the hybrid cell assembled with porous carbon electrodes. In order to confirm and expand this discussion, additional studies with ex situ X-ray photoelectron spectroscopy (XPS) and in situ Raman spectroscopy have been carried out. Furthermore, iodine electrodeposition has been investigated with electrochemical quartz crystal microgravimetry (EQCM) and transmission electron microscopy (TEM). The in situ Raman spectroscopy is used to monitor the carbon electrode surface through the electrolyte film. We show three stages of carbon material, from pristine to operating in a cell. The most important stage is the stable cycling, during which the nanoporous carbon electrode is reversibly taking up iodine in the form of solid I_2_. This leads to reversible p-doping of the carbon and mechanical strain, likely due to the formation of highly distorted solid I_2_ in the pores. However, before the stable stage can be reached, the electrode has to go through initial cycles of electrochemical polarization. During these first few cycles, highly reactive sites (such as defects) are still available that irreversibly react with the electrolyte to form C=O or C–I bonds. Even before electrochemical polarization, adsorption of iodine and minor surface oxidation are observed upon the first contact between the carbon and the electrolyte. To sum up, on the way to the desired stable cycling regime, the carbon surface in an iodine-based energy storage device is modified. Of these effects, the initial surface oxidation is irreversible, which is followed by reversible iodine adsorption, preventing further surface reactions. This might have significant implications for device performance given recent results regarding the dependence of iodine uptake on the surface oxidation in carbon [52].

## 2. Materials and Methods

### 2.1. EQCM Measurements

The mass and charge were measured during cyclic voltammetry with an EQCM 10 M combined with the Interface™ 1010 B Potentiostat/Galvanostat/ZRA (Gamry Instruments, Warminster, PA, USA). The microbalance was set for 5 MHz quartz crystals and the temperature controlled QCM Cell Kit (by Gamry Instruments) was used. All the experiments were conducted at room temperature. As a working electrode, the area of coated 5 MHz quartz was set at 1.1 cm^2^. The reference electrode was a Ag/AgCl 3 M KCl and as a counter electrode, a Pt wire was used. Spray coating was done with an airbrush-Set from Craftomat and the pressure used for spray coating was 2 bar.

A slurry with proper viscosity had to be produced in order to have a homogeneous and thin coating on the quartz electrode. BLACK PEARLS^®^ 2000 was selected for preparing the electrodes for EQCM, due to its small average particle size (15–20 nm) and high surface area, which favors the preparation of a reliable/stable electrode layer via spray coating [53]. BP 2000 Carbon Black (by Cabot corporation) possesses a surface area of SBET = 1450 m^2^ g^−1^ and a microporous volume of Vmi = 0.454 cm^3^ g^−1^. Together with polyvinylidene difluoride (PvDF) as binder and N-Methyl-2-pyrrolidone (NMP) as a solvent, a slurry was mixed to get the following composition: BP2000:PvDF:NMP = 9:1:174. First, the slurry was mixed for 20 min with a magnetic mixer and then 20 min with an ultrasonic disperser. The quartz crystals were spray-coated using a spray gun containing the slurry over the working area of the crystal. The quartz crystals were pre-heated using a magnetic mixer at 100 °C for two hours. To achieve a homogeneous coating, in total, four coating layers were applied on one crystal, each one separated by a 2-min
break between the second and the third layer. Taking into account the coating density of 0.4 g cm^−3^, the coating thickness between 10 μm and 15 μm and area mass loading of 0.3–0.4 mg are expected.

Further, in order to prevent the corrosion of the gold current collector, a relatively low concentration of 0.5 mol L^−1^ NaI was used for the EQCM investigations. It has been previously shown that gold corrosion can only occur if an oxidant such as iodine is present along with a high concentration of iodide ions in the electrolyte [54]. Therefore, the risk of corrosion in this experiment was mitigated by the use of a pristine NaI electrolyte. Furthermore, the electrochemical polarization during the EQCM measurements on the carbon electrode was limited to a potential range between 0.0 V and +0.5 V, on the one hand, to avoid corrosion at the positive polarity and on the other hand to properly monitor the mass changes during iodide/iodine redox reactions. Thus, the EQCM data gives accurate insight into the mass changes occurring inside the pores of the electrode without involving parasitic or additional reactions.

The measured resonance frequency change is compared to the calculated resonance frequency change Δ*f_cal_*. To calculate Δ*f_cal_*, the charge change Δ*Q* and the molecular mass *M* are taken as variables in Equation (1) (see also Appendix A) where F is the Faraday constant and C_f_ the Sauerbrey constant. During negative charging, the electrode accumulates positive ions and/or repels negative ions, while during positive charging, negative ions are attracted and/or positive ions repelled. Therefore, the negative sign of the formula is negated by a second negative sign. Thus, the charge number *z_i_* of the expected ion will determine whether Δ*f_cal_* will be directly or indirectly proportional to the charge. If an exchange mechanism is expected, the molecular mass of the solvated or non-solvated cation is subtracted by the molecular mass of iodide and the charge number *z_i_* equals 2.
(1)$$\Delta f_{cal}=\frac{\Delta Q\times C_{f}\times M}{F\times z_{i}}$$

Here, the calculated resonance frequency change using the suspected molecular mass is compared with the measured resonance frequency expected to be caused by the exchange mechanisms. In this case, the denoted molecular mass *M* is derived by the molecular mass of the solvated or non-solvated sodium cation *M*_sodium_ subtracted by the molecular mass of iodide *M*_iodide_ and divided by 2 as shown in Equation (2) (see also Appendix A).
(2)$$M=\frac{M_{sodium}-M_{iodide}}{2}$$

### 2.2. Electrodes for In Situ Raman Spectroscopy

All electrodes used for in situ Raman studies were prepared in free-standing form by mixing of 90 wt.% activated carbon (MSP-20, Kansai Coke and Chemicals), 5 wt.% carbon black (C65, Imerys), and 5 wt.% polytetrafluoroethylene (60% dispersion in water from 3 M Chemicals) in isopropanol. The mixture was stirred at 70 °C until the solvent was evaporated. The resulting dough was pressed and rolled with a custom-built calendaring machine to a thickness of 150 μm and dried at 120 °C to evaporate water and solvent adsorbed in the pores. The sodium iodide (Alfa-Aesar, Ward Hill, MA, USA) based aqueous electrolyte was used at a concentration of 1 mol L^−1^ with pH = 6.5 and 82 mS cm^−1^ conductivity. The diameter of the carbon electrodes was 10 mm, which were separated by a glassy fiber GF/A separator of a 260 μm thickness.

### 2.3. Raman Spectroscopy

Raman measurements were done using a LabRAM HR 800 spectrometer (Japan) combined with an Olympus BX41 microscope. All measurements were done using a 532 nm laser with a reduced power of 0.5 mW to avoid sample damage/alteration. The slit was set to 200 µm, the pinhole to 500 µm, and a 300 lines/mm grating was used. The acquisition time was 4 × 30 s for every spectrum (a total of 120 s/spectrum) and the DuoScan System was used to continuously scan the laser spot over a 20 × 20 µm area. The objective lens used was a 40× Olympus LUCPlanFLN (NA = 0.6; adjustable for different cover thicknesses). For the in situ measurements (during stable cycling) and the semi in situ measurement (first couple of cycles) a two-electrode in situ cell, ECC-Opto-Gas from EL-CELL (Hamburg, Germany), was used, and the measurements were done in backscatter geometry from the top. The mass ratio of the working to the counter electrode was 1:2, with the measurements done on the working electrode. The in situ measurements were done by continuously measuring Raman spectra while cycling the cell (0.08 mV s^−1^/−0.2–0.6 V) and after the cell had been cycled 5 times at a scan rate of 2 mV s^−1^ to reach the stable regime. For the semi-in situ measurements the cell was cycled separately (2 mV s^−1^/−0.2–0.6 V) and stopped at the end of every cycle to allow time for several Raman measurements at different positions (the spectra shown are averaged over those positions). The measurements of the first contact were done in a glass petri dish with the H_2_O or 1 M NaI/H_2_O added accordingly, and again an average over several measurement positions was done. The evaluation of the polyiodides-bands was done by deconvolution of the I_3_^−^/I_5_^−^ region using two Lorentz peaks (initial position 110 cm^−1^, 143 cm^−1^) and one Gaussian peak (initial position 165 cm^−1^), based on the assignment from [55]. The evaluation of the G- and D-bands was done as described by ref. [26]. 

### 2.4. TEM

Scanning Transmission Electron Microscopy (STEM) was performed on an aberration corrected FEI Titan G2 60–300, operated at 300 kV. The microscope was equipped with a SuperX EDX spectrometer. The sample preparation of a MSP20 electrode (cycled at 2 mV s^−1^/−0.2–0.6 V) and removed at 0.6 V was performed with ultramicrotomy after embedding in epoxy (Epofix, Struers, Germany). Ultrathin sections (<70 nm) were produced on the Leica Ultramicrotome UC6 (Leica Microsystems, Vienna, Austria) equipped with a 45° diamond knife (Diatome, Biel, Switzerland). After cutting, the thin sections were transferred to Quantifoil R3/3 TEM grids. Data acquisition and analysis were performed using the Gatan Digital Micrograph (GMS) Suite (v3.42, https://www.gatan.com/products/tem-analysis/gatan-microscopy-suite-software, accessed on: 3 April 2022). For determination of the iodine atomic positions the STEM HAADF, images were background corrected by subtracting a 128 times binned and subsequently rescaled copy of the corresponding.

### 2.5. XPS

X-ray photoelectron spectroscopy was performed in an ultrahigh vacuum chamber equipped with a dual anode X-ray source (SPECS XR 50, Berlin, Germany) and a hemispherical energy analyzer (SPECS Phoibos 150, Berlin, Germany). The carbon electrode samples were fixed on a Ti sample plate and transferred into the UHV chamber via a load-lock. Spectra were acquired with Al Kα radiation (400 W). Quantification was performed within the Prodigy software (SPECS v4.102.3, https://www.specs-group.com/, accessed on: 5 May 2022) and peak fitting was performed with XPSPEAK (v4.1, https://xpspeak.software.informer.com/4.1/, accessed on: 5 May 2022). The samples were cycled from −0.2 to 0.6 V (2 mV s^−1^) for 5 cycles before extracting the electrode.

## 3. Results and Discussion

### 3.1. EQCM Investigation of Iodine Electrodeposition in Porous Carbon Electrode

As a direct method for investigating the iodine deposition and behavior of water molecules and ionic species, in situ EQCM studies were performed. The resonance frequency changes observed in EQCM measurements can be translated into mass changes according to the Sauerbrey equation [56]. Previously, it has been established that an activated-carbon coating containing micropores with entrapped water is rigidly coupled to the underlying crystal, thus it serves as a true gravimetric probe of its microporous volume content [57]. During the first cyclic voltammogram (CV in Figure 1a), the positive sweep started from an open circuit potential of −10 mV. The resonance frequency is almost constant until iodide oxidation starts at ~0.3 V vs. Ag/AgCl leading to the onset of iodine formation. This region is followed by a steep increase of the resonance frequency, which is related to the oxidation of iodide to iodine in agreement with the oxidation potential of iodide (+0.54 V vs. standard hydrogen electrode). A significant increase in resonance frequency of 4772 Hz after the first cycle indicates a high mass loss, which could be due to the hydrophobic nature of iodine pushing out the water molecules and/or desorption of partially de-solvated Na^+^ ions to compensate the charge. A non-linearity of calculated versus experimental resonance frequency indicates the occurrence of chemical processes such as iodine/iodide comproportionation reactions (I_3_^−^ and I_5_^−^ formation) or due to the disturbed electrode/electrolyte coupling leading to a frequency rise. Here, the change in hydrophobicity of the carbon surface due to iodine formation affects the wetting of electrode. In addition, the iodine multilayer deposition-related mechanical stress in the nanopores (as discussed in the upcoming Raman section) could lead to a different response of two frequencies [58].

The full CV scan presented in Figure 1b between −0.5 V and +0.5 V is the second cycle, where a relatively small frequency change in the negative potential region (the EDL charging) is followed by a sharp increase in resonance frequency indicating the accumulation of iodine. To see the reversibility of the iodide/iodine redox reactions, in Figure 1c the CV is presented in the potential window from 0.0 V to +0.5 V where a decrease of resonance frequency until 0.2 V vs. Ag/AgCl can be related to the current changing direction from negative to positive that might correspond to the permeation selective Na^+^ ion adsorption. A reversible resonance frequency gain (mass loss) during the positive scan and a partially reversible resonance frequency decrease (mass gain) during the negative scan can be correlated to the high reversibility of the iodide/iodine reaction. However, the change in mass vs. charge (Figure 1d) is only partly reversible. The hysteresis of the mass change is attributed to side-reactions due to iodine/iodide comproportionation and surface reactions of carbon (discussed in the next sections). Overall, a frequency increase during the oxidation of iodide and a decrease during the reduction of iodine are caused by the transport of a considerable amount of water, which moves out or into the pores with Na^+^ ions and iodine deposition-related reversible chemo-mechanical processes.

### 3.2. Structural Parameters of Carbon during Stable Cycling

We begin this section with an interesting question from a device-engineering point of view: Are there any modifications of the carbon structure during regular cycling? If so, what are these changes and the extent of their reversibility?

Using Raman measurements, the carbon electrode can be closely monitored during the cycling operation. Here, the carbon electrode composition and the electrolyte concentration differ from the EQCM studies due to the different cell constructions (see the Section 2). The top row of Figure 2a–c shows one exemplary cycle with Raman spectra extracted during the polyiodides’ formation. In the region between 100 and 200 cm^−1^, the formation of polyiodides can be observed with I_3_^−^ and I_5_^−^ generating at different rates, the details of which were discussed earlier [1]. In addition, we can see subtle changes in the carbon bands that correlate with the amount of polyiodides, both in the extracted spectra in Figure 2b as well as when comparing the intensity plots in Figure 2c,f. Figure 2d shows the averaged Raman spectra of the carbon electrode (over 0.1 V steps) over one exemplary cycle during the I_3_^−^ and I_5_^−^ formation (red spectra) and decomposition (blue spectra), whereas the spectra at the same voltage are nearly identical, revealing no hysteresis between formation and decomposition. In order to reveal clearer trends, a combined analysis of several cycles, measurement positions on the electrodes and across different cells, has been realized.

Figure 3 shows the changes to the band parameters of the G- and D-bands (position, width, intensity ratio), where the deconvolution was done according to the works of Ferrari & Robertson [26], as a function of the intensity ratio of the I_3_^−^-band (110 cm^−1^) to the G-band, which serves as a semi-quantitative measure of the number of polyiodides formed. Note that these figures look qualitatively the same if the intensity of the I_5_^−^-band is used instead of the I_3_^−^-band (Appendix A). In this analysis, clear trends become visible, as all band parameters change approximately linearly with the number of polyiodides. The D-band appears to be affected more strongly with both an upshift and a narrowing of several cm^−1^. In addition, the I_D_/I_G_ ratio decreases significantly by up to −0.1. Based on a wide array of studies on the influence of charge transfer and iodine/polyiodide adsorption on carbon [6,7,8,9,10,11,12,13,14,15,16,17,18,19,20,21,22], we attribute these changes mainly to the p-doping of carbon, in agreement with earlier reports [23,24,25,52]. In line with the earlier (ex situ) work of our group, the changes to the D-band are larger and clearer than those of the G-band [24,25]. It is not obvious why this is the case, but one needs to consider that fundamental studies on charge transfer effects are usually done on highly regular carbon substrates, such as graphite [39], graphene [40,42,45,46,47,48,49,50,51], or carbon nanotubes [43,44]. More amorphous carbon, which is used in most practical devices, might behave significantly differently with regard to the defect-related D-band. In addition, the interpretation of the G-band behavior is further complicated by changes induced by the pressure inside the carbon pores during cycling, which is on the same order of magnitude as the charge transfer related changes.

For the G-band, we observe a very small narrowing and downshift. Whilst the narrowing of the G-band is qualitatively in line with the charge transfer interpretation, though much smaller than that of the D-band, an upshift rather than a downshift of the G-band is expected. Here, the electrodeposition of solid iodine, which is elaborated with EQCM and TEM experiments, seems to play a role. We believe that the iodine forming in the highly constrained conditions of the carbon pores exerts pressure on the carbon. Therefore, the downshift of the G-band is attributed to the strain caused by this pressure. More precisely, the total shift of the G-band is a superposition of an upshift due to charge transfer and a downshift due to the mechanical strain, with a net effect of a slight downshift. In addition, in the case of the G-band, a stress-induced band splitting may appear as a broadening of the G-band, which means that the changes in the band-width are very small due to the effects of charge transfer and mechanical strain pointing in the opposite direction.

This interpretation of the G-band behavior was further investigated by measurements of the distribution of iodine atoms in the polarized carbon electrode using aberration corrected STEM high-angle annular dark-field (HAADF) imaging. The contrast in this imaging mode is dominated by Rutherford scattering and therefore, the obtained signal is roughly proportional to the square of the atomic number [61], which makes it ideal for visualizing heavier elements in a lighter matrix. Figure 4a depicts an overview of a microtome cut of the electrode suspended over the TEM support grid. Since iodine is the element with the highest atomic number present in the sample, we can assign the bright features in the atomically resolved HAADF images to iodine in Figure 4b,c. The iodine atoms are visible as bright spots that are superimposed over a thickness-dependent grey background caused by the carbon matrix. Note that the often blurry or elongated appearance of the atoms results from their movement under the electron beam [62,63]. Some of the atoms are part of small clusters composed of approximately three to twelve atoms, which roughly corresponds to the number of I-atoms that would fit into large mesopores of MSP-20. Representative examples of such agglomerations are marked with circles in Figure 4c. The EDX analysis shown in Figure 4d provides clear evidence for the presence of iodine.

To assess the stochastic distribution and interatomic distances of the iodine atoms, we automatically determined the atom positions by a supervised learning approach based on a Random Forest classifier (Figure 4e) [64]. It is possible to identify 586 atoms in the image shown in Figure 4e, corresponding to an areal density of approximately 1 atom/nm^2^. From the obtained positions, we also calculated the radial distribution function (RDF) given in Figure 4f. In the RDF, two main peaks are identified at distances of 2.7 Å and 3.8 Å, which agree well with the formation of a highly distorted I_2_-nanocrystal found by in situ SAXS in earlier work [1] and which is also qualitatively consistent with the interpretation of the G-band behavior in Raman data. However, these values represent upper limits for the corresponding real interatomic distances in the sample, because they are measured in projection. In principle, it is also possible that two or more atoms are aligned parallel to the electron beam and are therefore detected as a single atom. However, a quantitative histogram analysis of the feature intensities, inset in Figure 4f, shows a unimodal distribution, indicating that most of the detected features are indeed single atoms.

To sum up this part, p-doping of the carbon by iodine/polyiodides and the mechanical strain of the carbon have been observed, due to the solid iodine formation in the nanopores. The other important question related to reversibility of changes should be addressed from the point of view of the carbon, the iodine, and the polyiodides. The reversible polyiodides formation/decomposition is obvious from the Raman data (Figure 1). The reversibility of the carbon modifications can also be checked from the Raman data. In order to make sure that even small changes would be noticeable, the Raman spectra have been extracted at the beginning and end of the cycles for every measurement position (Appendix A). The spectra are nearly identical in all cases and the average changes for all parameters are negligible between cycles (Appendix A). We thus conclude that after the initial surface reactions are completed, the carbon remains stable during cycling with only reversible changes occurring due to charge transfer to iodine/polyiodides and mechanical strain due to the solid iodine formation. This also indirectly points to a fully reversible iodine deposition.

### 3.3. Structural and Surface Properties during Initial Cycles

From the foregoing, it is clear that the carbon structure is eventually stabilized and the modifications that occur during one cycle are reversible. However, an initial oxidation does occur, during which irreversible changes happen by surface oxidation and reactions with iodine. This means that the surface of the pristine carbon is different from the one effectively used during normal cell operations. We have studied the surface reactions with a semi-in situ Raman approach supported by XPS measurements. The Raman spectra and band parameters of the same electrode as a function of cycles are summarized in Figure 5a–f. The clearest changes are a broadening of both the G- and D-bands that saturate somewhere between the 3rd and 5th cycle. Band broadening is usually attributed to a decrease in structural order and/or sp^2^-content [26,27]. In addition, we see a constant I_D_/I_G_ ratio, a nearly constant G-band position, and a slight upshift of the D-band. This is inconsistent with the classic behavior of carbon amorphization [26,27]. However, it could be consistent with surface oxidation [65,66,67,68,69], if one considers the fairly amorphous starting point of this carbon compared to the examples reported in the literature. We thus assign these changes to surface oxidation and possible reactions with iodine at the reactive defect sites on the highly irregular carbon surface that saturates over time.

To check this interpretation, XPS measurements were carried out on the pristine carbon after its immersion in the electrolyte (starting condition of the Raman measurement) and two electrodes after cycling. The results are summarized in Figure 5g,h, where the C1s’ peak is separated into contributions from sp^2^, sp^3^, C–O, C=O and C–I. In line with the Raman results, we see little difference between the pristine material and the material after immersion in the electrolyte, as well as an increase in C=O and C–I bonds and a decrease of sp^2^ and sp^3^ for the electrodes after cycling. Also, signals for O and I increase based on the quantification of the XPS results (Appendix A). We thus conclude that during the first couple of cycles, the carbon is irreversibly modified by surface oxidation and reactions with iodine. The extent of the oxidation of the carbon surface might have implications for the amount of iodine-uptake during operation, as has recently been demonstrated on a graphene oxide-based system [56]. Two points are noteworthy here: Firstly, in any real-world device, it is the modified carbon surface that is relevant for the lifetime of the device. Secondly, in laboratory settings, one needs to keep this initial reaction phase in mind, in order to avoid contradictory results that could arise, for example, from comparing results from the very first cycles to the end of the cycling.

The question arises at this point if there is a similar irreversible modification already upon the first contact with the electrolyte. This is especially relevant to the interpretation of ex situ experiments, where the pristine materials are often characterized before any contact with the electrolyte.

### 3.4. Structural and Surface Properties at Simple Immersion of Carbon in the Electrolyte

The influence of the first physical contact between the carbon surface and iodide electrolyte should be evaluated to confirm the origin of changes in electrode parameters. Therefore, we would like to briefly consider what is happening before the “zeroth” cycle. That is, are there any modifications of the carbon upon the first contact with the electrolyte?

Raman spectroscopy allows comparing the parameters of carbon under various conditions. Figure 6a shows the pristine material in the air, immersed in H_2_O and in 1 M NaI/H_2_O electrolyte. Figure 6b shows again the pristine material, but this time compared to the sample dried after immersion in H_2_O and in 1 M NaI/H_2_O electrolyte and the sample rinsed in H_2_O for about 1 min after immersion in a 1 M NaI/H_2_O electrolyte. Figure 6a shows the total changes due to the contact with the electrolyte, whilst Figure 6b shows the irreversible changes. For the immersion in water, the changes appear to be minor, with the only clear change being a decrease in the I_D_/I_G_ ratio and a slight upshift of the D-band. The G-band position appears unaffected and both the G- and D-bands broaden slightly, though this is close to the limit of accuracy of our measurement (Figure 6c–g). We attribute this to the adsorption of water molecules on the carbon surface with an additional increase of disorder due to minor surface oxidation, as described in the case of the first few cycles. The majority of the changes appear to be due to the adsorption of water molecules, as upon drying, the changes are almost fully reversed (Figure 6b–g), with only a slight decrease of the I_D_/I_G_ ratio (Figure 6g). The XPS measurements (Figure 6h,i) also show minor changes in the C–O and C=O contributions.

During immersion in the 1 M NaI/H_2_O electrolyte, we see a larger decrease in the I_D_/I_G_ ratio, band broadening of both bands, and an upshift of both bands. In addition to the effects observed in water, we attribute this mainly to the adsorption of I- and a subsequent charge transfer from the carbon (upshift of both bands + decrease in I_D_/I_G_). According to a theoretical study of iodine on carbon by Tristant et al. [8] most of the I^−^ likely reacts to form I_2_ on the surface, though some I^−^ species might be found on defect sites as well. In their work, both I^−^ and I_2_ caused p-doping of carbon. The band broadening, in that case, could be due to band splitting, which has been described for the adsorption of I_2_ vapor on graphene [10]. Such an effect would appear as a broadening in this case, due to the naturally large band width compared to highly ordered graphene. Interestingly, these changes are only partially reversed upon drying the carbon. This is due to the fact that iodine remains on the surface, as is evident from the XPS results (Appendix A). However, most of the iodine is present in the form of polyiodides, which is visible in the dried sample (Appendix A). These polyiodides can subsequently be removed by washing the sample in H_2_O (Figure 6), making the carbon revert back to its original state. The remaining, irreversible changes (decrease in I_D_/I_G_, small broadening of both bands) are nearly similar to what occurred with the immersion in H_2_O and are again attributed to minor surface oxidation. The XPS results are also comparable between the immersed sample in H_2_O, and the one in 1 M NaI/H_2_O and its subsequent washing with H_2_O (Figure 6h).

Overall, some irreversible surface oxidation similar to what happens during the first couple of electrochemical cycles occurs upon first contact with the electrolyte. However, the overall effects are negligible. At the same time, an initial charge transfer between adsorbed I^−^/I_2_ and the carbon occurs that is only reversible if the iodine is washed off after removing the sample from the electrolyte.

## 4. Conclusions

The electrodeposition of iodine in nanoporous carbon delivers a low-cost cathode with fast charge/discharge properties and a detailed understanding of the carbon-iodine interface is central to the development of this technology. Both the carbon electrode surface and structural parameters are important from the ageing point of view. Iodine electrodeposition inside the pores (in aqueous NaI) drives an enormous number of water molecules along with partially de-solvated Na^+^ cations out of the porous carbon electrode. Nevertheless, the iodine accumulation puts a mechanical strain on the carbon electrode, which is favorably counter-balanced by the charge transfer pointing in the opposite direction. Thereby the net effect of iodine on carbon structural parameters is highly reversible and restored with each cycle and reduces the ageing risks during long-term cycling.

However, the surface interactions of the aqueous iodide electrolyte may not be prevented completely. The carbon electrode in iodine/carbon-based energy storage first undergoes irreversible surface modifications before reaching a stable regime after about five cycles. The carbon surface oxidation and carbon-iodine reactions start with the initial contact with the aqueous NaI, accelerate with electrochemical polarization and eventually reach a steady state. At this point, a passivation of the electrode takes place due to iodine consuming the majority of the reactive sites (or dangling bonds) and the electrode enters into a stable cycling regime.

The carbon modification during initial cycles has two important implications. Firstly, when doing ex situ experiments, particular care must be taken in comparing pristine material with the treated electrodes, as a mere contact with the electrolyte can modify the electrode surface. Secondly, the carbon electrodes, during most of the energy storage devices lifetime, will have different surface chemistry than the pristine material. Furthermore, it is demonstrated that the nanoporous carbon undergoes small structural changes during the stable cycling regime, due to a charge transfer (p-doping) by solid iodine and polyiodides, as well as a mechanical strain by distorted nanocrystals of iodine forming in the carbon pores. These changes are found to be reversible in the stable cycling regime and should not influence the long-term performance of energy storage devices.

## Figures and Tables

**Figure 1 nanomaterials-13-01545-f001:**
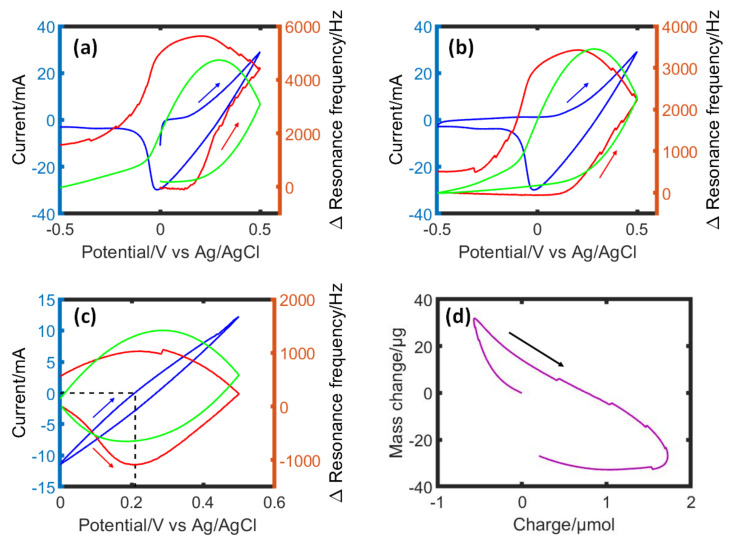
Electrochemical quartz crystal microgravimetry (EQCM) investigation of a BP2000 carbon electrode in 0.5 mol L^−1^ NaI. (**a**) The first cycle of cyclic voltammetry (CV) at a scan rate of 10 mV s^−1^ starting from an open circuit potential of −10 mV to +0.5 V and then to −0.5 V, (**b**) second cycle from −0.5 V to +0.5 V and (**c**) a representative CV cycle from 0.0 V to +0.5 V polarization versus Ag/AgCl reference potential. The measured resonance frequency (red) is compared with the calculated one (green) based on Equations (1) and (2) (Section 2). The current is shown in blue. The working electrode is a carbon coated quartz crystal, the counter electrode is a platinum wire, and the reference is an Ag/AgCl electrode. The calculated resonance frequency change is based on the permeation selective cation charging mechanism according to a molecular mass change per charge of sodium Na^+^ × H_2_O (41 g mol^−1^, sodium adsorbs and desorbs in nanometric pores of carbon at partially de-solvated state with one water molecules) [59,60]. (**d**) Mass change versus charge change for the data in (**c**).

**Figure 2 nanomaterials-13-01545-f002:**
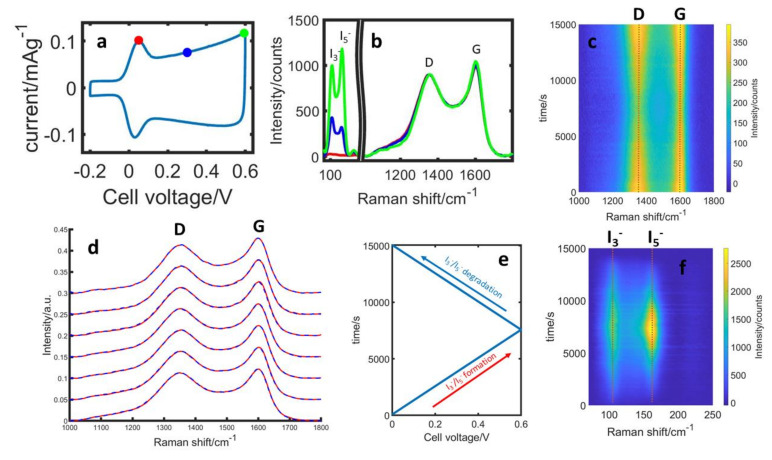
Exemplary in situ Raman results from one charge/discharge cycle during stable cycling. (**a**) Current-voltage curve from one cycle (scan rate 0.08 mV s^−1^). (**b**) Raman spectra of the polyiodides and carbon region extracted at the beginning (red), middle (blue) and end (green) of the I_3_^−^/I_5_^−^ formation, the black like indicates axis break. (**c**) Intensity of the G and D band as a function of time during the voltage scan. (**d**) Normalized Raman spectra of the carbon at various cell voltages (example of one cycle) during I_3_^−^/I_5_^−^ formation (red) and I_3_^−^/I_5_^−^ decomposition (blue). The spectrum at 0.6 V is the same data in both cases. (**e**) Cell voltage as a function of time. (**f**) Intensity of the I_3_^−^ (110 cm^−1^) and I_5_^−^ (165 cm^−1^) bands as a function of time.

**Figure 3 nanomaterials-13-01545-f003:**
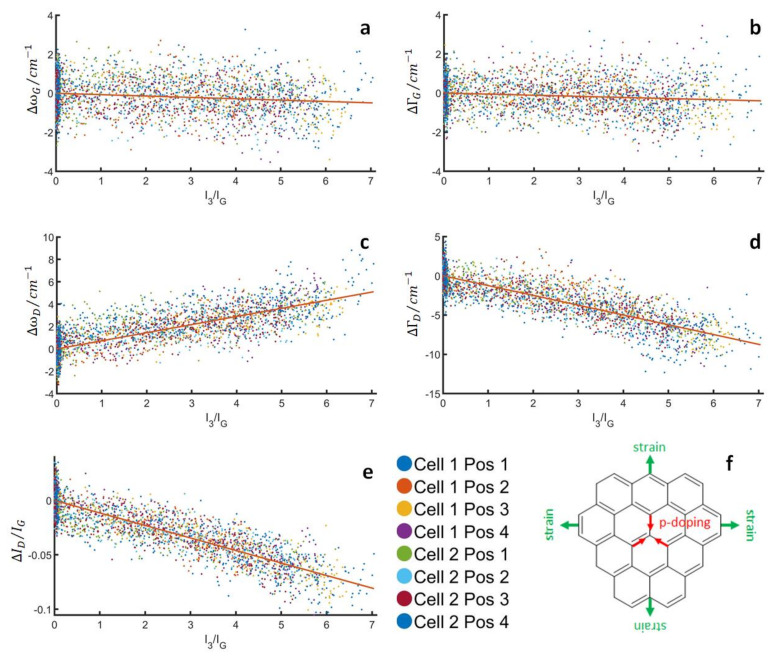
Changes in the G- and D-band parameters as polyiodides are generated at the carbon electrode. The intensity ratio of the main I_3_^−^ band (110 cm^−1^) and the G-band (1600 cm^−1^) are used as a measure for the amount of polyiodides formed. (**a**) G-band position, (**b**) G-band width, (**c**) D-band position, (**d**) D-band width (**e**) intensity ratio of the D-/G-band and (**f**) sketch of the opposite effects of mechanical strain and p-doping in the carbon structure.

**Figure 4 nanomaterials-13-01545-f004:**
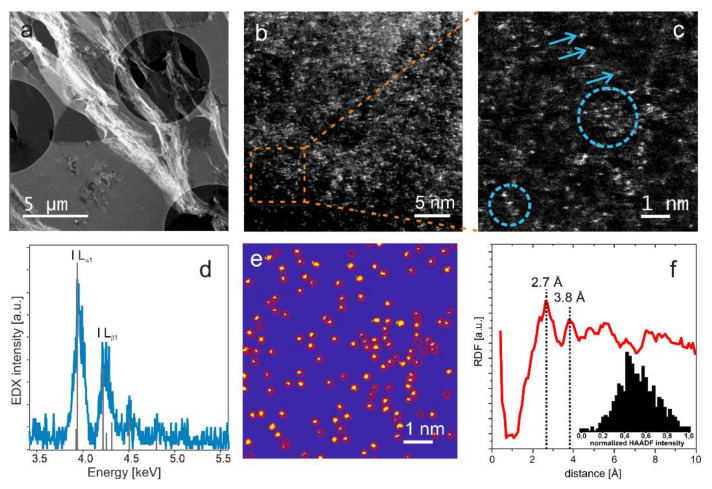
TEM analysis: (**a**) STEM HAADF overview image of the microtome cut suspended over the TEM grid. (**b**) High-resolution STEM HAADF micrograph showing the distribution of iodine atoms in the carbon matrix. (**c**) magnified region marked in (**b**). Iodine atoms and clusters are exemplary, highlighted by arrows and circles, respectively. (**d**) EDX spectrum demonstrating the presence of iodine. (**e**) result of the automatic segmentation procedure of the region depicted in (**c**). (**f**) radial distribution function calculated from obtained atom positions in (**e**). The histogram in the inset represents a quantitative analysis of the normalized intensities corresponding to the detected features.

**Figure 5 nanomaterials-13-01545-f005:**
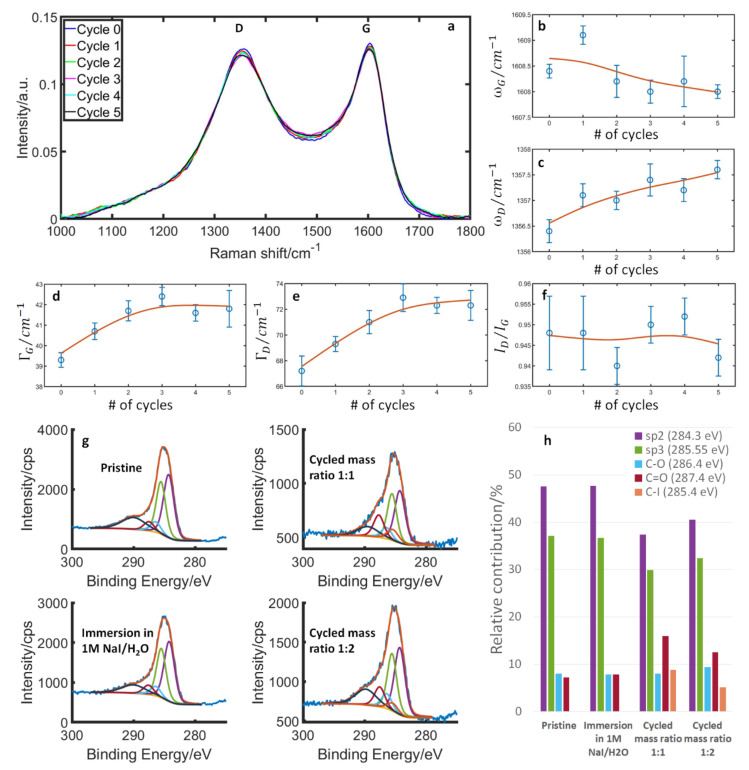
Raman and XPS measurements of the carbon electrode during initial cycles. (**a**) Raman spectra from the same MSP 20 electrode measured at a cell voltage of 0 V after 0–5 cycles, the measurements are done in the in situ cell. (**b**–**f**) Band parameters of the G- and D-band as a function of the numbers of cycles, the red lines are a smoothing function to guide the eye: (**b**) G-band position, (**c**) D-band position, (**d**) G-band width, (**e**) D-band width and (**f**) intensity ratio of the D-/G-band. (**g**) XPS spectra of the C1s’ region of the pristine material, after immersion in 1 M NaI/H_2_O and after cycling with a counter electrode mass ratio of 1:1 and 1:2. The cycling was done from −0.2 to 0.6 V (scan rate = 2 mV s^−1^). The carbon peak is separated into contributions from sp^2^, sp^3^, C–O, C=O and C–I. (**h**) The relative contributions of these are shown in the bar diagram.

**Figure 6 nanomaterials-13-01545-f006:**
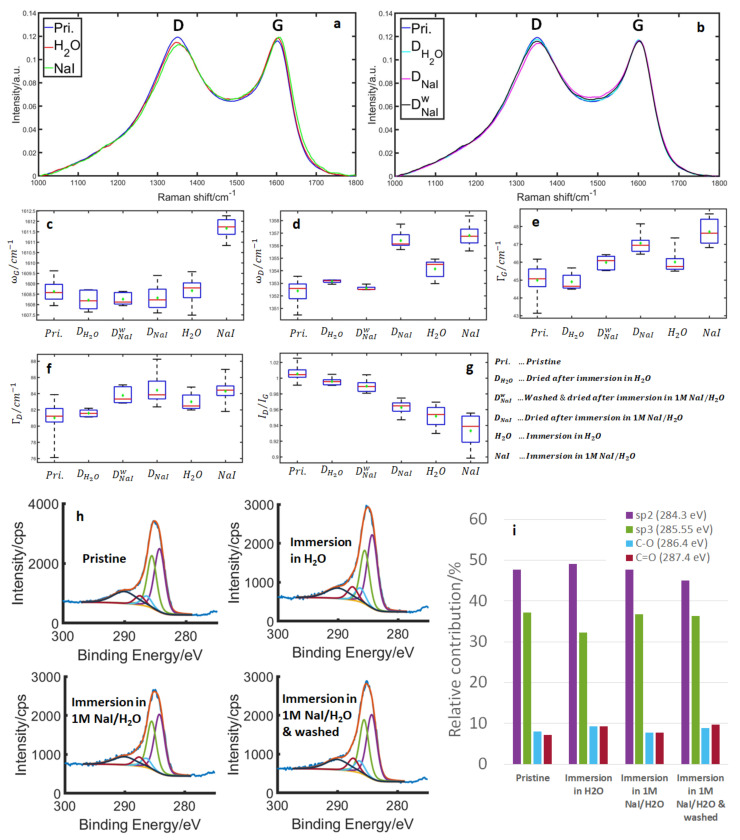
(**a**) Raman spectra of pristine MSP20 compared to MSP 20 immersed in H_2_O and MSP 20 in a 1 M NaI/H_2_O electrolyte. (**b**) Raman spectra of pristine MSP 20 compared to MSP 20 dried after immersion in H_2_O, MSP 20 dried after immersion in a 1 M NaI/H_2_O electrolyte and MSP 20 washed (in H_2_O) and dried after immersion in a 1 M NaI/H_2_O electrolyte. (**c**–**g**) Boxplots of the band parameter of the G- and D-bands for all tested conditions, note that the red line marks the median and the green diamond the average: (**c**) G-band position, (**d**) D-band position, (**e**) G-band width, (**d**) D-band width and (**g**) intensity ratio of the D-/G-band. (**h**) XPS spectra of the C1s’ region of the pristine material, after immersion in H_2_O, after immersion in 1M NaI/H_2_O and after immersion in 1M NaI/H_2_O and subsequent washing in H_2_O. (**i**) The carbon peak is separated into contributions from sp^2^, sp^3^, C–O, C=O. The relative contributions of these are shown in the bar diagram on the right.

## Data Availability

The data presented in this study are available on request from the corresponding author.

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
