# Peer review of "Impact of Iodine Electrodeposition on Nanoporous Carbon Electrode Determined by EQCM, XPS and In Situ Raman Spectroscopy"

_nanomaterials, 2023, doi:10.3390/nano13091545_

Round 1
Reviewer 1 Report
The authors have performed an elaborate work to develop deep understanding of iodine electrodeposition and have used various metrologies to explain the observations and have correlated the results obtained via different methods. Overall, the story is well written and on the level of scientific interpretation a ‘strong’ impression is left behind. The work can be accepted after minor changes, which are listed below. Some of them are typos and a few suggestions are included, as well.
Line 104
[achieve an homogeneous] > [achieve a homogeneous]
Line 111
[if an oxidants such as iodine] > [if an oxidant such as iodine] or [if oxidants such as iodine]
Equation 1 + 2 on line 132 + 140: please put them in an equation editor, to enhance the readability / resolution.
Line 136
[molecular mass of the solvated or non-solvated sodium cation] The authors could consider elaborating on this statement. For instance, something along the lines of the text in the caption of figure 1. Or at least add a few references at this point, to support this.
Line 150
Check correct writing of unit [mS cm-1]
Line 182
[is equipped] > [was equipped]
Line 220 + 242
Check correct writing of [Na+]
Line 222
Check correct writing of [(I3- and I5- formation)]
Line 239-240
[surface reactions of carbon] > Can the authors elaborate on which type of reactions ? Is this known from literature or by their own experimental work ? Refer to the Raman section ?
Line 281+282
[formation (red spectra) and decomposition 281 (blue spectra)]
I understand what the authors are referring to, however, this is slightly confusing. In figure 2D, I do not see ‘red’ or ‘blue’ spectra and in figure 2E there are no spectra shown (only a ‘red’ and ‘blue’ line indicating at which time scale the formation / decomposition occur. Please consider to rewrite this in a more clear fashion.
Line 281
[y cycle during the I3 - and I5 – formation] The authors should either include the chemical reactions that explain the formation of both species from the sodium iodide solution or include a reference in which this can be found.
Line 282
[whereas the spectra at the same voltage are nearly identical]
I find this somewhat difficult to see. This is referring to figure 2F ? Indeed, the colour ‘cloud’ seems similar in the formation and decomposition part of the figure, however, this is not too clear. Perhaps change figure 2D to a ‘flat’ figure, without 3D perspective and limit the number of spectra to (1) formation spectrum and (2) decomposition spectrum.
Figure 2D: cell voltage goes up to 0.6 V and figure 2E: cell voltage goes up to 0.5 V
Figure 2B: the authors could consider to remove that black wavy line that indicates the axis break. Much clearer and less overwhelming to have an axis break indicated at the axis and not all over the figure. When break signs are used at the top + bottom X axes, that should be sufficiently clear.
Line 355
Check correct writing of [atom/nm2]
Line 357
[of 2.7 A and 3.8 A] I suppose that is Angstrom ? Correct symbol to be used.
Line 383
[oxidation do occur] > [oxidation does occur]
Line 464
Check correct writing of [I-]
Line 497
[reduces the of ageing risks] check English formulation
Author Response
Response to Reviewer #1:
The authors have performed an elaborate work to develop deep understanding of iodine electrodeposition and have used various metrologies to explain the observations and have correlated the results obtained via different methods. Overall, the story is well written and on the level of scientific interpretation a ‘strong’ impression is left behind. The work can be accepted after minor changes, which are listed below. Some of them are typos and a few suggestions are included, as well.
Line 104
[achieve an homogeneous] > [achieve a homogeneous]
Reply: Corrected
Line 111
[if an oxidants such as iodine] > [if an oxidant such as iodine] or [if oxidants such as iodine]
Reply: Corrected
Equation 1 + 2 on line 132 + 140: please put them in an equation editor, to enhance the readability / resolution.
Reply: Corrected
Line 136
[molecular mass of the solvated or non-solvated sodium cation] The authors could consider elaborating on this statement. For instance, something along the lines of the text in the caption of figure 1. Or at least add a few references at this point, to support this.
Reply: As it is elaborated together with references in the caption of Fig.1, we would like not to repeat the same in the main text.
Line 150
Check correct writing of unit [mS cm-1]
Reply: Corrected
Line 182
[is equipped] > [was equipped]
Reply: Corrected
Line 220 + 242
Check correct writing of [Na+]
Reply: Corrected
Line 222
Check correct writing of [(I3- and I5- formation)]
Reply: Corrected
Line 239-240
[surface reactions of carbon] > Can the authors elaborate on which type of reactions ? Is this known from literature or by their own experimental work ? Refer to the Raman section ?
Reply: The oxidation of carbon surface is well-known in aqueous electrolytes, please refer to references in XPS section 65-69
Line 281+282
[formation (red spectra) and decomposition 281 (blue spectra)]
I understand what the authors are referring to, however, this is slightly confusing. In figure 2D, I do not see ‘red’ or ‘blue’ spectra and in figure 2E there are no spectra shown (only a ‘red’ and ‘blue’ line indicating at which time scale the formation / decomposition occur. Please consider to rewrite this in a more clear fashion.
Thank you for pointing out this confusing figure/sentence combination. The sentence is (fully) referring to Figure 2d and has been changed accordingly.
Figure 2d shows the averaged Raman spectra of the carbon electrode (over 0.1 V steps) over one exemplary cycle during the I3- and I5- formation (red spectra) and decomposition (blue spectra), whereas the spectra at the same voltage are nearly identical, revealing no hysteresis between formation and decomposition.
Line 281
[y cycle during the I3 - and I5 – formation] The authors should either include the chemical reactions that explain the formation of both species from the sodium iodide solution or include a reference in which this can be found.
No change was made here as a reference to a paper on the details of the I3-/I5- formation is made in the same paragraph two sentences before (Line 275: In the region between 100-200 cm-1, the formation of polyiodides can be observed with I3- and I5- generating at different rates, the details of which have been discussed earlier [1].) Referencing the same paper again seems excessive.
Line 282
[whereas the spectra at the same voltage are nearly identical]
I find this somewhat difficult to see. This is referring to figure 2F ? Indeed, the colour ‘cloud’ seems similar in the formation and decomposition part of the figure, however, this is not too clear. Perhaps change figure 2D to a ‘flat’ figure, without 3D perspective and limit the number of spectra to (1) formation spectrum and (2) decomposition spectrum.
Again thank you for pointing out the ambiguity in the description here. As already mention the sentence is about Figure 2d and was changed accordingly. In our opinion, it is important to show several spectra during formation and degradation to point out both the changes to the spectra (qualitatively) at different voltages and the reversibility at each voltages step. However, we have turned it into a 2D figure, which hopefully makes it easier to read and interpret.
Figure 2D: cell voltage goes up to 0.6 V and figure 2E: cell voltage goes up to 0.5 V
Thank you for this input. In fact both in both figure 2d and 2e (and 2a) the cell voltage goes up to 0.6 V, but figure 2E had a confusingly ticked x-axis. Figure 2E has been changed to make this clearer.
Figure 2B: the authors could consider to remove that black wavy line that indicates the axis break. Much clearer and less overwhelming to have an axis break indicated at the axis and not all over the figure. When break signs are used at the top + bottom X axes, that should be sufficiently clear.
We thank the reviewer for this input, but would like to keep the axis break as is.
Line 355
Check correct writing of [atom/nm2]
Reply: Corrected
Line 357
[of 2.7 A and 3.8 A] I suppose that is Angstrom ? Correct symbol to be used.
Reply: Corrected
Line 383
[oxidation do occur] > [oxidation does occur]
Reply: Corrected
Line 464
Check correct writing of [I-]
Reply: Corrected
Line 497
[reduces the of ageing risks] check English formulation
Reply: Corrected

Reviewer 2 Report
This manuscript investigates the impact of iodine storage on the physical and chemical characteristics of the carbon host in aqueous iodide electrolytes. It is proposed that interactions at the carbon/iodine interface occur in two stages, beginning with irreversible surface changes and progressing to a stable regime. In the first few cycles, the highly reactive sites on the electrode surface are irreversibly modified by surface oxidation and reaction with iodine to generate C-I and C=O bonds, and these irreversible reactions will restrict the formation of subsequent irreversible surface reactions. During the stable cycle stage, the reversible conversion of iodine ions to iodine happens in the nanopores of carbon materials, resulting in reversible p-doping and mechanical strain of carbon materials. In general, this manuscript presents an interesting area of research and details the carbon-iodine interactions at each stage. This is a meaningful and relatively complete work. Here are some questions that the author should address further:
1. In Figure 1a, according to the description in the manuscript (at line 215), a steep increase of the resonance frequency at 0.54V (vs SHE) is related to the oxidation of iodide to iodine. However, in the process of oxidation of iodine ions to iodine, the total mass of the electrode material should increase, the corresponding frequency should decrease. How can this paradoxical behavior be explained?
2. Why was the peak of solid iodine not represented in the Raman data when analyzing the influence of iodine deposition on the structure of carbon materials?
3. Please provide the XPS data of the C1s, O1s and I3d after multiple cycles to demonstrate that the irreversible reactions are effectively inhibited in subsequent long cycles.
4. It is mentioned that the iodine deposition process on carbon materials includes irreversible reactions in the first few cycles and stable reversible reactions in future cycles. Please give the relevant charge/discharge and cycle data.
Author Response
Response to Reviewer #2:
This manuscript investigates the impact of iodine storage on the physical and chemical characteristics of the carbon host in aqueous iodide electrolytes. It is proposed that interactions at the carbon/iodine interface occur in two stages, beginning with irreversible surface changes and progressing to a stable regime. In the first few cycles, the highly reactive sites on the electrode surface are irreversibly modified by surface oxidation and reaction with iodine to generate C-I and C=O bonds, and these irreversible reactions will restrict the formation of subsequent irreversible surface reactions. During the stable cycle stage, the reversible conversion of iodine ions to iodine happens in the nanopores of carbon materials, resulting in reversible p-doping and mechanical strain of carbon materials. In general, this manuscript presents an interesting area of research and details the carbon-iodine interactions at each stage. This is a meaningful and relatively complete work. Here are some questions that the author should address further:
- In Figure 1a, according to the description in the manuscript (at line 215), a steep increase of the resonance frequency at 0.54V (vs SHE) is related to the oxidation of iodide to iodine. However, in the process of oxidation of iodine ions to iodine, the total mass of the electrode material should increase, the corresponding frequency should decrease. How can this paradoxical behavior be explained?
Reply: Thank you for the comment. As explained in the text related with Fig.1, the oxidation of iodide leads to iodine formation. Although one would think, the electrode gets heavier after iodine is deposited and that corresponding EQCM response should indicate a mass gain. It is rather contrary to this perception. The resonance frequency increases during oxidation of iodides (formation of iodine). This is essentially due to the hydrophobic nature of iodine which expels a lot of water from the nanopores of carbon. In addition, since the iodides are consumed, the sodium ions together with their solvation shells move out of the pores as well. Both these expulsions from the porous electrode lead to a net mass loss which is why the resonance frequency keeps increasing as long as the oxidation of iodides proceeds.
- Why was the peak of solid iodine not represented in the Raman data when analyzing the influence of iodine deposition on the structure of carbon materials?
Reply: The oxidation of iodide to iodine is not directly indicated in Raman spectra. We could not find the iodine directly. However, we could well-establish based on our previous Raman findings that polyiodides (I3- and I5-) are indicated during charging which can only form from iodine. This is proven in our previous studies via in situ Raman spectroscopy and SAXS methods (please ref to reference 1 of the manuscript).
- Please provide the XPS data of the C1s, O1s and I3d after multiple cycles to demonstrate that the irreversible reactions are effectively inhibited in subsequent long cycles.
Reply: We regret that we do not have the XPS data after several cycles at the moment. We are planning new long-term experiments which will be part of our next manuscripts.
- It is mentioned that the iodine deposition process on carbon materials includes irreversible reactions in the first few cycles and stable reversible reactions in future cycles. Please give the relevant charge/discharge and cycle data.
Reply: Please see Fig S8.

Round 2
Reviewer 2 Report
I suggest the acceptance of the revised manuscript